# Molecular Mechanism of Autosomal Recessive Long QT-Syndrome 1 without Deafness

**DOI:** 10.3390/ijms22031112

**Published:** 2021-01-23

**Authors:** Annemarie Oertli, Susanne Rinné, Robin Moss, Stefan Kääb, Gunnar Seemann, Britt-Maria Beckmann, Niels Decher

**Affiliations:** 1Institute for Physiology and Pathophysiology, Vegetative Physiology, Philipps-University of Marburg, 35037 Marburg, Germany; annemariekuhn@web.de (A.O.); rinne@staff.uni-marburg.de (S.R.); 2Institute for Experimental Cardiovascular Medicine, University Heart Center Freiburg–Bad Krozingen, Medical Center-University of Freiburg, 79110 Freiburg, Germany; kai.robin.moss@universitaets-herzzentrum.de (R.M.); gunnar.seemann@universitaets-herzzentrum.de (G.S.); 3Faculty of Medicine, University of Freiburg, 79110 Freiburg, Germany; 4Department of Medicine I, University Hospital, LMU Munich, 80336 Munich, Germany; Stefan.Kaab@med.uni-muenchen.de (S.K.); beckmann@med.uni-frankfurt.de (B.-M.B.); 5Deutsches Zentrum für Herz-Kreislauferkrankungen (DZHK), Partner Site Munich, 80636 Munich, Germany; 6Institut für Rechtsmedizin, Forensische Molekularpathologie, Universitätsklinikum Frankfurt, 60590 Frankfurt, Germany

**Keywords:** KCNQ1, LQTS, potassium channel, electrophysiology

## Abstract

*KCNQ1* encodes the voltage-gated potassium (Kv) channel KCNQ1, also known as KvLQT1 or Kv7.1. Together with its ß-subunit KCNE1, also denoted as minK, this channel generates the slowly activating cardiac delayed rectifier current *I*_Ks_, which is a key regulator of the heart rate dependent adaptation of the cardiac action potential duration (APD). Loss-of-function mutations in *KCNQ1* cause congenital long QT1 (LQT1) syndrome, characterized by a delayed cardiac repolarization and a prolonged QT interval in the surface electrocardiogram. Autosomal dominant loss-of-function mutations in *KCNQ1* result in long QT syndrome, called Romano–Ward Syndrome (RWS), while autosomal recessive mutations lead to Jervell and Lange-Nielsen syndrome (JLNS), associated with deafness. Here, we identified a homozygous *KCNQ1* mutation, c.1892_1893insC (p.P631fs*20), in a patient with an isolated LQT syndrome (LQTS) without hearing loss. Nevertheless, the inheritance trait is autosomal recessive, with heterozygous family members being asymptomatic. The results of the electrophysiological characterization of the mutant, using voltage-clamp recordings in *Xenopus laevis* oocytes, are in agreement with an autosomal recessive disorder, since the *I*_Ks_ reduction was only observed in homomeric mutants, but not in heteromeric *I*_Ks_ channel complexes containing wild-type channel subunits. We found that KCNE1 rescues the KCNQ1 loss-of-function in mutant *I*_Ks_ channel complexes when they contain wild-type KCNQ1 subunits, as found in the heterozygous state. Action potential modellings confirmed that the recessive c.1892_1893insC LQT1 mutation only affects the APD of homozygous mutation carriers. Thus, our study provides the molecular mechanism for an atypical autosomal recessive LQT trait that lacks hearing impairment.

## 1. Introduction

There are two important potassium currents responsible for the late phase of repolarization of the cardiac action potential (AP): the rapid delayed rectifier potassium current (*I*_Kr_) and the slow rectifier potassium current (*I*_Ks_) [1,2]. The stimulation of beta-adrenergic receptors in the heart increases heart rate and contractility and shortens the duration of the AP by activating *I*_Ks_ [3]. The *I*_Ks_ current is generated by KCNQ1 channels, which co-assemble with their cardiac ß-subunit KCNE1, also known as minK [4]. KCNQ1 channels, also denoted as Kv7.1, belong to the family of voltage-dependent potassium (Kv) channels which contain six transmembrane segments, with cytoplasmic N- and C-termini [5]. Four α-subunits assemble to a tetrameric structure with a pore-loop located between the S5 and S6 segments [6] that contain the selectivity filter of the channel [7]. In contrast, the *KCNE1* gene encodes for a single transmembrane domain with an extracellular N-terminus and a cytoplasmic C-terminal end. All KCNE family members (KCNE1-5) assemble with KCNQ1 and modify the channel characteristics [8]. In the heart, KCNE1 is the primary accessory subunit of KCNQ1 [9], causing drastic changes to the electrophysiological characteristics of the KCNQ1 channel: current amplitudes are increased, activation is delayed and voltage-dependence of activation is shifted to more positive potentials [4,5]. In addition, KCNE1 is essential for the targeting of KCNQ1 to the surface membrane and its respective stability in the plasma membrane. This becomes evident as in *MinK* (*KCNE1*) knock-out mice, no KCNQ1 immunostaining of the luminal membrane of dark cells of the vestibular system was observed [10]. KCNE1 mutations were identified to be clinically relevant in the human heart and inner ear [10].

Besides the heart, KCNQ1 is expressed in several tissues including kidney, pancreas, lung, placenta, colon, spleen, prostate, peripheral blood leukocytes, small intestine and the stria vascularis of the cochlear duct [11,12]. *KCNQ1* knockout mice presented symptoms such as deafness, balance problems, and morphological anomalies in the internal ear and in the gastrointestinal tract [13]. The KCNQ1/KCNE channel complex is highly relevant for the physiology of the inner ear and a variety of epithelial tissues [14]. For instance, KCNQ1 and KCNE1 were detected in the apical membranes of the marginal cells of the stria vascularis and the dark cells of the vestibular end-organs [15,16,17].

Congenital long QT syndrome (LQTS) is an inherited channelopathy characterized by a prolonged QT-interval and arrhythmias caused by delayed repolarization [18]. The prevalence of LQTS is 1:2000 in apparently healthy newborns [19]. The clinical manifestations of LQTS are divided into two main categories: arrhythmias and electrocardiogram (ECG) alterations. The cardiac events are due to runs of torsades de pointes ventricular tachycardias, eventually leading to syncope or even to cardiac arrest and sudden cardiac death. *KCNQ1* mutations are the most common with a rate of 39–49% [20,21,22]. Depending on the type and location of the *KCNQ1* mutation, it can cause rare autosomal recessive Jervell and Lange-Nielsen Syndrome (JLNS) or autosomal dominant Long QT Syndrome 1, also denoted as Romano–Ward Syndrome (RWS) (a less severe form) [23,24]. Homozygous or biallelic compound mutations in *KCNQ1* or *KCNE1* genes were initially reported to lead to JLNS, which is characterized by congenital deafness combined with syncopal attacks and sudden death due to prolonged frequency corrected QT interval (QTc) [25,26]. In the inner ear, *I*_Ks_ maintains the homeostasis of potassium in the endolymph. The inner ear seems to tolerate *I*_Ks_ channel deficits much better than the heart. Thus, it was shown that only KCNQ1 protein levels lower than 10% of the normal level lead to JLNS [23]. Patients with autosomal recessive LQT1 do not suffer from hearing loss. Genetically, they show homozygous or biallelic missense compound mutations [24]. Patients with autosomal recessive LQT1 and JLNS patients present a high rate of cardiac events. The lack of hearing loss in autosomal recessive LQT1 patients is not accompanied by an improved prognosis [24]. *MinK* knock-out mice display inner ear defects with deafness similar to what is observed in patients with JLNS, again highlighting the relevance of the KCNQ1/KCNE1 channel complex in the inner ear [27]. Lastly, autosomal dominant LQT1, known as RWS, is a clinically mild form with genetically heterozygous missense, nonsense, exon skipping and frameshift mutations [23]. However, some KCNQ1 variants can cause both JLNS (in an autosomal recessive manner) and RWS (in an autosomal dominant manner) [28]. The extant of the loss-of-function determines the symptoms of a *KCNQ1* mutation carrier. A complete loss-of-function results in JLNS and, according to the proportion of the remaining functional KCNQ1 channels, autosomal recessive LQT1 or autosomal dominant LQT1 is developed [23].

The present study focuses on the identification of the molecular mechanisms of the *KCNQ1* variant, p.P631fs*20 (c.1892_1893insC), leading to rare autosomal recessive LQTS without hearing impairment. The identical mutation was first reported in 1999 by Neyroud et al. [29], and a few years later, Novotny et al. published a case report about a 7-year-old boy suffering from recurrent syncopes and carrying this variant in a homozygous manner [30]. However, previous electrophysiological characterization did not reveal why the mutation is only detrimental in the homozygous state. To this end, we performed voltage-clamp experiments in *Xenopus laevis* oocytes expressing the variant alone or together with wild-type KCNQ1, in the absence and presence of the cardiac *I*_Ks_ subunit KCNE1. Here, we found that KCNE1 rescues the trafficking defect of the heterozygous complex of wild-type and mutant KCNQ1 channel complexes, while it cannot rescue the impaired function of the homomeric p.P631fs*20 (c.1892_1893insC) mutant channel complex.

## 2. Results

### 2.1. Identification of a Homozygous KCNQ1 Gene Mutation in a Family with LQTS without Hearing Impairment

Since the age of three, the index patient (IP) (PID 1889) suffered from syncopes, triggered by exercise, sleep or QT prolonging medication. Based on these symptoms, combined with a repeated prolonged QTc interval (Figure 1A) (560 ms under β-blocker-therapy) and a positive family history, we diagnosed LQTS. Notably, the IP did not suffer from hearing loss and she was also diagnosed with multiple sclerosis. To probe for inherited LQTS, we used a candidate gene approach and identified a *KCNQ1* variant. The insertion of cytidine (c.1892_1893insC) leads to a frameshift starting from amino acid P631 and a premature stop codon at position 651 (p.P631fs*20, denoted as KCNQ1^P631fs*20^) (Figure 1B,C); KCNQ1^WT^ comprises 676 amino acids. The frameshift in the KCNQ1 amino acid sequence is located in the cytoplasmic C-terminus of the KCNQ1 channel protein (Figure 1C). The variant was absent from the Exome Variant Server (EVS) (https://evs.gs.washington.edu/EVS/) and not found by the gnomAD database (former ExAC browser) (https://gnomad.broadinstitute.org/). Referring to the Protein Variation Effect Analyzer (PROVEAN)-score (http://provean.jcvi.org/index.php), KCNQ1^P631f^^s*20^ was predicted to be deleterious.

The IP was homozygous for the *KCNQ1* c.1892_1893insC variant and both parents were heterozygous and asymptomatic mutation carriers (Figure 1D). Additionally, one paternal uncle (QTc 383 ms) and a paternal aunt (QTc 412 ms) were heterozygous for the mutation. The sister died at the age of 21 because of sudden cardiac death (SCD), and she was, in contrast to the IP, deaf. Her brother died when he was 2 years old due to SCD. Two daughters and a son of the IP were heterozygous carriers of the mutation and asymptomatic (QTc 421 ms, 397 ms and 413 ms, respectively) (Figure 1D).

### 2.2. Electrophysiological Characterization of the KCNQ1^P631fs*20^ Variant Revealed a Loss-of-Function in Both Homozygous and Heterozygous states

To characterize the electrophysiological properties of KCNQ1^P631fs*20^, the variant was expressed in *Xenopus laevis* oocytes and compared to oocytes injected with KCNQ1 wild-type (KCNQ1^WT^) copy RNA (cRNA) using voltage-clamp recordings. We observed a significant reduction in the current amplitude for the homomeric variant (resembling the homozygous state) (Figure 2A–C). Analyzing the outward currents at +40 mV, the current amplitude was reduced by about 65% compared to KCNQ1^WT^ (resembling healthy controls) (Figure 2C). Injecting similar amounts of KCNQ1^WT^ and KCNQ1^P631fs*20^ (resembling the heterozygous state) led to a significant current reduction (Figure 2A–C); however, it was less pronounced than observed for the variant expressed alone. Current reduction analyzed at +40 mV was about 40% compared to the KCNQ1^WT^ (Figure 2C). We also injected half the amount of KCNQ1^WT^ cRNA to mimic a haploinsufficiency, and found currents comparable to the heterozygous state (Figure 2A–C). Thus, heteromeric channels that contain the mutation suffer from a non-dominant negative but clear loss-of-function.

### 2.3. In Electrophysiological Recordings, KCNE1 Rescued the KCNQ1^P631fs*20^ Loss-of-Function Present in the Heteromeric State with KCNQ1^WT^

Since in cardiac tissue KCNQ1 is expressed in a complex with its ß-subunit KCNE1 to form the *I*_Ks_, we also examined the electrophysiology of the KCNQ1^P631fs*20^ mutant co-expressed with KCNE1. As *Xenopus laevis* oocytes express an endogenous KCNQ1 channel (*x*KCNQ1), we first recorded the “endogenous *xI*_Ks_” by only injecting KCNE1 (Figure 3A) to determine the *x*KCNQ1/KCNE1 background for the subsequent analysis of the exogenously expressed hKCNQ1/KCNE1 channel complex. Here, we observed a significant reduction in the current amplitude for KCNQ1^P631fs*20^ injected together with KCNE1 (resembling the homozygous state) compared to KCNQ1^WT^ plus KCNE1 (Figure 3A–C). The outward current analyzed at +40 mV was reduced by about 65% (Figure 3C). The current amplitude was comparable to that of the endogenous *xI*_Ks_ background. Surprisingly, injecting similar amounts of KCNQ1^WT^ and KCNQ1^P631fs*20^ together with KCNE1 (resembling the heterozygous state) led to no current reduction (Figure 3A–C), in contrast to the current reduction observed for the heterozygous state without KCNE1 (Figure 2A–C). As a control, we also injected half the amount of KCNQ1^WT^ cRNA together with KCNE1 to mimic a haploinsufficiency (Figure 3A–C), and found that the current reduction was less pronounced, with only 35% at +40 mV compared to KCNQ1^WT^ expressed with KCNE1 (healthy control) (Figure 3C). In summary, the electrophysiological data indicate that there are strongly reduced *I*_Ks_ currents for homozygous KCNQ1^P631fs*20^ carriers as observed in our IP, and no *I*_Ks_ current reduction for heterozygous mutation carriers. Thus, KCNQ1^P631fs*20^ channels assembled without KCNE1 also display a current reduction in the heterozygous state (Figure 2), which is rescued by KCNE1 (Figure 3). Since, in the heart, KCNQ1 is predominantly assembled together with KCNE1, these results may explain the LQTS and symptoms of the IP, and the absence of a phenotype of heterozygous mutation carriers.

### 2.4. The KCNQ1^P631fs*20^ Mutant does Not alter the Gating of Homomeric and Heteromeric I_Ks_ Channel Complexes

Next, we analyzed whether the KCNQ1^P631fs*20^ mutation affects the voltage-dependence of activation of *I*_Ks_ channel complexes containing KCNE1 (Figure 4). The voltage-dependence of the homomeric channel complex of KCNQ1^P631fs*20^ and KCNE1 was similar to that of the human or *Xenopus* wild-type *I*_Ks_ channels (Figure 4A), with no significant changes in the voltage of half-maximal activation (V_1/2_) (Figure 4B). Additionally, the “KCNE1 rescued” heteromeric *I*_Ks_ channel complexes, containing KCNQ1^WT^, KCNQ1^P631fs*20^ and KCNE1 subunits, did not have a significantly altered voltage-dependence (Figure 4A,B), indicating that the loss-of-function and the rescue of the KCNQ1^P631fs*20^ currents primarily results from changes in the trafficking of the *I*_Ks_ channel complex by KCNE1.

### 2.5. Action Potential Modelling Predicts a QT Prolongation Exclusively Present In Homozygous Patients

Action potential simulations were performed using the O’Hara-Rudy ventricular cell model [31] after the subtraction of the endogenous *xI*_Ks_ current, recorded by the injection of KCNE1 alone (Figure 3C and Figure 5A). Thus, simulations were carried out with a completely reduced conductivity of the *I*_Ks_ channel for the homozygous state and with an 11% reduction for the heterozygous state (Figure 5A). For the homozygous mutation carriers, the simulations predicted a prolongation of the action potential duration (APD). In contrast, the simulations predicted that the APD of the heterozygous patients was not or only very mildly prolonged (Figure 5B), resulting in APD_90_ values of 232 ms for homozygous patients and 222 ms for heterozygous patients (APD_90_ for wild type was 221 ms), respectively (Figure 5C). Taken together, the computational simulations confirmed that the APD, and thus the QT prolongation, occurs in homozygous, but not in heterozygous mutation carriers.

## 3. Discussion

In the present study, we describe the inheritance of the *KCNQ1* mutation p.P631fs*20 (c.1892_1893insC) in a family with homozygous and heterozygous mutation carriers over three generations. The homozygous IP presented a pronounced prolongation of the QT interval (QTc 560 ms) with several syncopal events, but without hearing loss. The identical mutation was first reported in 1999 by Neyroud et al. [29]. Here, the heterozygous IP experienced several stress-induced syncopes since the age of three, and her QTc interval was 450 ms [29]. Despite the fact that the father of the IP shared the heterozygous mutation, he was asymptomatic. Nevertheless, the authors proposed RWS, meaning an autosomal dominant inheritance. Our data clearly indicate that the heterozygous state does not cause a loss-of-function or act in a dominant-negative manner, as typically observed for RWS mutations. Most importantly, no further genetic analysis of the family was described in the study by Neyroud et al., thus an additional mutation responsible for the symptoms of the heterozygous patient cannot be excluded.

A few years later, Novotny et al. published a case of a 7-year-old boy suffering from recurrent syncopes and a prolonged QTc interval (up to 700 ms). This boy carried the variant we analyzed in our study in a homozygous manner [30]. The heterozygous mutation carriers of the family of our IP (PID 1889) were asymptomatic, which is in line with the family reported by Novotny et al., where the heterozygous parents of the young boy were also asymptomatic. Both our IP as well as the boy in this case report had no hearing loss. Novotny et al. proposed recessive LQT1 syndrome due to KCNQ1^P631fs*20^. However, in the study of Neyroud et al., as well as in the study by Novotny et al., no electrophysiological characterization of the mutation was performed. Finally, Sato et al. presented a family with two heterozygous mutations, insC1893-1894 (P631fs/19) (named P631fs/20 in our study) and delGTA1783-1785 (delV595), causing autosomal-recessive compound heterozygous LQT1 [32]. The patients carrying the two compound mutations suffered from syncopes and had no hearing loss. The heterozygous family members carrying one of the *KCNQ1* mutations (insC1893-1894 (P631fs/19) or delGTA1783-1785 (delV595)) showed no significant QT prolongation, implying that both mutations lacked a dominant negative effect. This is in line with the lack of phenotype in the heterozygous family members of our IP. In whole cell patch clamp recordings, both mutations (P631fs/19 or delV595) showed a strong current reduction when expressed alone with KCNE1 (homozygous state); however, in the heterozygous state, meaning the expression of each mutant together with KCNQ1^WT^ and KCNE1, the current was not significantly altered [32]. Expressing both mutants together with KCNE1 (as in the compound heterozygous patients), current was strongly reduced [32]. Sato et al. did not perform electrophysiological experiments without KCNE1, and thus the authors could not find the loss-of-function in the heteromeric state, which is present in the absence of KCNE1, and the rescue of heteromeric channel complexes composed of mutant and wild-type channel subunits. As the compound heterozygous patients described by Sato et al. carried the mutation we identified, together with a second mutation, we speculate that KCNE1 can only rescue heteromeric channels containing wild-type KCNQ1 channel subunits.

The study from Sato et al. demonstrated a decreased expression of KCNQ1^P631fs*19^ channels caused by intracellular aggregation and trafficking defects. The new amino acid sequence resulting from the frame shift contains two potential newly generated endoplasmic reticulum (ER) retention signals (R^633^GR and R^646^LR). Thus, the authors proposed a retention of KCNQ1^P631fs*19^ in the ER [32]. Thus, the question arises, why can the mistrafficking of the channel and the associated current reduction be exclusively found in the homozygous state? KCNQ1^P631fs*19^ mutants still form heteromeric complexes with KCNQ1^WT^ subunits, but they did not efficiently retain the KCNQ1^WT^ subunits [32]. On the other hand, Sato et al. described that the KCNQ1^P631fs*19^ mutation severely affected the expression and stability of full-length KCNQ1 protein in transfected cells, and proposed, based on these observations, that the number of mutant KCNQ1 subunits was fewer than the number of KCNQ1^WT^ subunits in the ER, where the subunit assembly occurred. Thus, the amount of functional heteromeric channels at the plasma membrane may not be altered. However, in this study, KCNE1 was not co-transfected studying the trafficking and heteromerization of KCNQ1 channels. Our data indicate that KCNE1 is able to fully restore the mistrafficking of heteromeric complexes of wild-type and mutant channels since we did not observe a current reduction for the heterozygous state (in the presence of KCNE1). We did not perform any cell surface expression or trafficking experiments for KCNQ1^P631fs*20^ and KCNQ1^WT^ (with and without KCNE1), as the mutation has been previously described as trafficking deficient [32]. Since KCNE1 rescues the loss-of-function caused by a mistrafficking of this mutant, while the resulting *I*_Ks_ channel complex has a normal voltage-dependent gating, the most straightforward explanation is that it rescues the trafficking phenotype, especially as KCNE1 is known to modulate the surface expression of KCNQ1 [10,33,34,35]. Yet, as we did not obtain any primary data to confirm this theory, the mechanism of a rescued trafficking defect is currently only a hypothesis.

Bianchi et al. described a *KCNQ1* mutation (c.1630_1636delCAGTACTinsGTTGAGAT), denoted as Δ544, which is a deletion/insertion mutation leading to a novel amino acid sequence and a premature stop codon at position 651 [36]. The homozygous mutation carrier of Δ544 suffered from JLNS and the heterozygous mutation carriers presented a mild phenotype [29]. In voltage-clamp recordings of the mutated *I*_Ks_ current (co-expression of homomeric KCNQ1^Δ544^ and KCNE1), Bianchi et al. showed a current reduction down to the level of endogenous *xI*_Ks_ (injection of KCNE1 only) [36]. These data are comparable to our findings with homomeric KCNQ1^P631fs*20^ co-expressed with KCNE1, while our patient did not suffer from hearing loss. In contrast to our data with the KCNQ1^P631fs*20^ variant, the co-expression of KCNQ1^WT^ and KCNQ1^Δ544^ (resembling the heterozygous state) in the presence of KCNE1 led to a current reduction of about 50%, albeit not in a dominant negative manner. We did not observe such a reduction in *I*_Ks_ for KCNQ1^P631fs*20^ in the heterozygous state when KCNE1 is present. These differences may explain the mild phenotype of the heterozygous KCNQ1^Δ544^ mutation carriers and the absence of symptoms in heterozygous KCNQ1^P631fs*20^ carriers. Both mutations, KCNQ1^P631fs*20^ as well as KCNQ1^Δ544^, share a premature termination of the amino acid chain at the terminal C-terminus and present a severe phenotype in a homozygous manner. Interestingly, KCNQ1^Δ544^ presents, as well as KCNQ1^P631fs*20^, retention motifs due to the deletion/insertion. However, Bianchi et al. showed the expression of KCNQ1^Δ544^ at the plasma membrane, thus ER retention may not be the mechanism for the current reduction observed for KCNQ1^Δ544^, whereas a localization of the mutant in the ER and adjacent organelle, presumably the Golgi apparatus, was described for KCNQ1^P631fs*20^ [32]. Whether these mechanistic differences may explain the different clinical spectrum of LQTS in these families currently remains unclear. Our IP and other homozygous KCNQ1^P631fs*20^ mutation carriers previously described do not suffer from hearing loss. In our voltage-clamp experiments, homomeric KCNQ1^P631fs*20^ was expressed together with KCNE1, as it is the case in the inner ear and the heart of our IP. Bhuiyan et al. postulated that a minimum of functional channels prevents hearing loss [23]. The remaining *I*_Ks_ current may maintain the potassium homeostasis in the endolymph, ensuring proper inner ear function, while it may not be sufficient for normal electric activity of the heart. Why the inner ear is not or only very rarely affected, such as in the case of the sister of our IP, despite the massive current reduction that we have observed in the *Xenopus* oocyte expression system, remains unclear. The trafficking defect of the KCNQ1^P631fs*20^ mutation may manifest to a different extent depending on the tissue.

Our data are in agreement with an autosomal recessive inheritance trait of the KCNQ1^P631fs*20^ mutation. The mechanism behind this unusual recessive genetic LQTS trait only became obvious by co-expression studies in the absence and presence of KCNE1. Expressing the heterozygous KCNQ1^P631fs*20^/KCNQ1^WT^ channel complex without KCNE1 led to a current reduction of about 39%, comparable to a haploinsufficiency. This loss-of-function in the heterozygous state was completely rescued by KCNE1, while the homomeric mutant channel complexes were not recovered. Recessive inheritance can be explained by a KCNE1 mediated rescue, which is selective for heteromeric channels that contain wild-type and mutant channel subunits as found in heterozygous patients.

## 4. Materials and Methods

### 4.1. Clinical Evaluation

In 2003, the *KCNQ1* mutation p.P631fs*20 (c.1892_1893insC) was identified in the Index patients (IP) at the age of 39 years. The diagnosis of LQTS was set by cardiologists specialized on inherited arrhythmia syndromes at the Hospital of the Ludwig Maximilians University of Munich. Detailed medical history focused on LQTS relevant comorbidities and the family history for 3 generations. Additionally, 12-lead resting ECGs were taken. According to Goldenberg et al. [37], we considered a QT interval corrected for the heart rate using Bazett’s formula (QTc) ≤450 ms in men and ≤460 ms in women to be normal. Informed consent prior to genetic investigations was collected for all patients in this study, complying with the ethical standards of the 1964 Declaration of Helsinki and its latest revision at that time.

### 4.2. Molecular Biology

Blood samples of the IP were taken and genomic DNA was extracted. All exons and intronic splice sites of the genes *KCNQ1*, *KCNH2*, *KCNE1*, *KCNE2* and *SCN5A* were PCR amplified (primer sequences available upon request) and sequenced. For *KCNQ1*, obtained DNA sequence information was compared with the *KCNQ1* wild-type sequence (NM_000218.3). Human KCNQ1 was cloned into the pSP64T vector. QuikChange Site-Directed Mutagenesis Kit (Agilent Technologies, Santa Clara, CA, USA) was used to introduce the mutation into human KCNQ1 cDNA. cDNA was linearized with EcoRI (Thermo Fisher Scientifics, Waltham, MA, USA) and cRNA was in vitro synthesized using the mMESSAGEmMACHINE^®^SP6 kit (Thermo Fisher Scientifics, Waltham, MA, USA). cRNA concentration was quantified using a spectrophotometer (NanoDrop, Thermo Fisher Scientifics, Waltham, MA, USA) and quality was controlled by agarose gel electrophoresis.

### 4.3. Electrophysiology

cRNA was injected into *Xenopus laevis* oocytes. Oocytes were taken from ovarian lobes of anesthetized *Xenopus laevis* toads. Anesthesia was performed with 2 g/l tricaine-methansulfonate (SIGMA-Aldrich, Missouri, USA). Subsequently, oocytes were treated with collagenase type II (2 mg/mL, Worthington, USA) in OR2 solution (NaCl 82.5 mM, KCl 2 mM, MgCl_2_ 1 mM, HEPES 5 mM, pH 7.4; all from SIGMA-Aldrich, Missouri, USA) for 120 min to remove residual connective tissue. Isolated oocytes were stored at 18 °C in ND96 recording solution (NaCl 96 mM, KCl 2 mM, CaCl_2_ 1.8 mM, MgCl_2_ 1 mM, HEPES 5 mM, pH 7.5) supplemented with Na-pyruvate (275 mg/L), theophylline (90 mg/L) and gentamicin (50 mg/L) (all from SIGMA-Aldrich, Missouri, USA). Oocytes were injected with 50.6 nl of KCNQ1 cRNA or KCNQ1 and KCNE1 cRNA in equal parts. Standard two electrode voltage-clamp experiments were performed at room temperature (21–22 °C) with an Axoclamp 900A amplifier, a Digidata 1440A and pClamp10 software (Axon instruments, CA, USA) 72 h after cRNA injection. The microelectrodes were made from glass pipettes (Science products, Hofheim, Germany) pulled with a DMZ-Universal puller (Zeitz, Germany) and had a resistance of 0.2–1.0 MΩ when filled with 3 M KCl (SIGMA-Aldrich, Missouri, USA). The following voltage protocols were used: 3000 ms pulses were applied in 20 mV steps from −60 mV to +60 mV, or 7000 ms pulses were applied in 20 mV steps from −40 mV to +40 mV for recording oocytes expressing the KCNQ1 channels, or KCNQ1 and the KCNE1 subunit, respectively.

### 4.4. Action Potential Modelling

Action potential simulations were obtained using the O’Hara-Rudy ventricular cell model [31] in its endocardial configuration. The set of differential equations was solved with a fixed time step of 10 μs using the Rush-Larsen method [38] for all gating variables and the forward Euler method for all other variables at a pacing frequency of 1Hz. The conductivity of the gks channel was then reduced by 100% for the homozygous state and 11% for the heterozygous state, based on the respective measured mean current amplitudes. To ensure all configurations had reached limit cycle, 500 beats were simulated. APD_90_ was then calculated based on the last beat.

### 4.5. Data Analysis

All electrophysiology data were acquired with Clampex (Molecular Devices, Sunnyvale, CA, USA) and analyzed with Clampfit 10 (Molecular Devices, Sunnyvale, CA, USA), Origin (OriginLab Corp, Northampton, MS, USA) and Excel (Microsoft Corp, Seattle, WA, USA). All values are expressed as means ± S.E.M. Error bars in all figures represent S.E.M. values. Significance was assessed using two tailed Student´s t-test. Asterisks indicate significance: *, *p* < 0.05; **, *p* < 0.01; ***, *p* < 0.001.

## Figures and Tables

**Figure 1 ijms-22-01112-f001:**
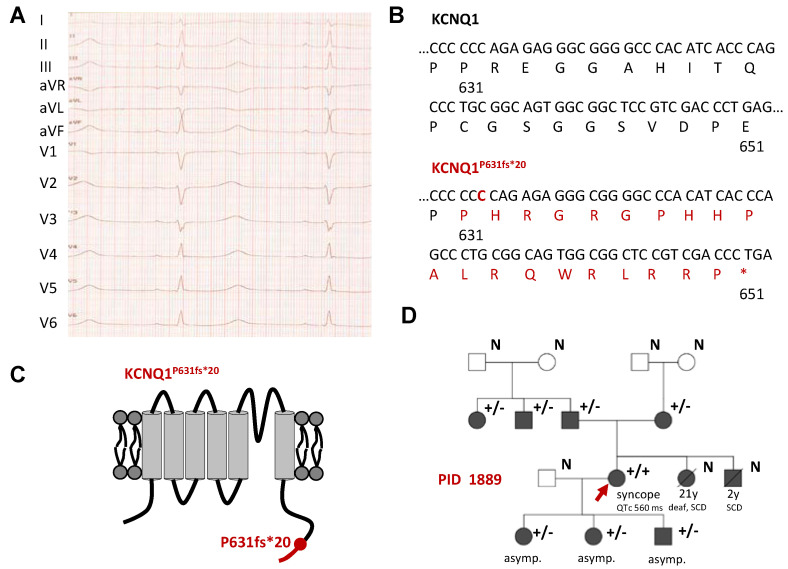
*Identification of the homozygous KCNQ1^P631fs*20^ mutation in a patient with LQTS without hearing loss.* (**A**) 12-lead ECG recording of the IP (PID1889) showing a QTc of 560 ms and a heart rate of 60 bpm under beta-blocker therapy. Paper speed was 50 mm/sec. (**B**) Partial nucleotide and amino acid sequence illustrated from amino acid 630 to 651 for KCNQ1^WT^ (top) and the KCNQ1^P631fs*20^ variant (bottom). The cytidine insertion is highlighted in red as well as the novel amino acid sequence due to the frame shift. Please note that the mutated sequence shows a premature stop codon at amino acid position 651. (**C**) Cartoon showing the topology of the KCNQ1 channel α-subunit. The location of the mutation is indicated by a red circle and the resulting novel amino acid sequence due to the frameshift is provided in red. (**D**) Pedigree of the family of the IP (marked by a red arrow) with a KCNQ1^P631fs*20^ mutation as a result of a nucleotide insertion mutation (c.1892_1893insC). Filled symbols indicate patients and family members with a previous diagnosis of LQTS with or without symptoms. Squares and circles represent male and female subjects, respectively. In the top right of the symbol, genetic information is given: “+/−” heterozygous mutation carrier, “+/+“ homozygous mutation carrier, “N” no genetic information was available. Below the symbols, information about certain symptomatic or further diseases are given. “SCD” sudden cardiac death. In addition, the QTc is shown for the IP. If there are no symptoms, it was marked with “asymp.”. Symbols with a line through mark deceased subjects, and the age and cause of death is indicated below.

**Figure 2 ijms-22-01112-f002:**
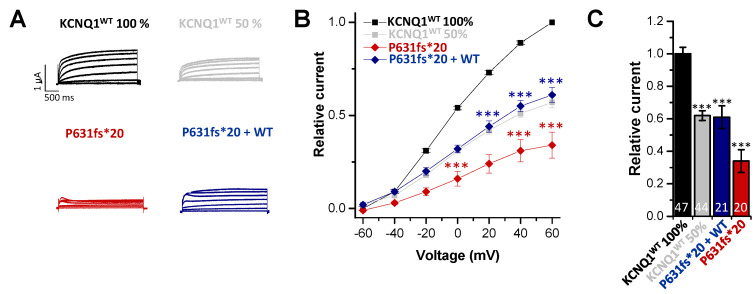
*Functional effects of the KCNQ1^P631fs*20^ mutation.* (**A**) Representative current traces of oocytes injected with KCNQ1^WT^ 100% (14.5 ng/oocyte), KCNQ1^P631fs*20^ (14.5 ng/oocyte), KCNQ1^WT^ 50% (7.25 ng/oocyte) or KCNQ1^WT^ plus KCNQ1^P631fs*20^ (7.25 ng/oocyte each), respectively. Voltage was stepped from -60 to +60 mV in 20 mV steps lasting 3000 ms, from a holding potential of −80 mV. (**B**) Current voltage relationships obtained by blotting the current at the end of each voltage step for each voltage applied normalized to KCNQ1^WT^. In order to obtain the current-voltage relationship (I/V curve), all wild-type recordings were normalized to the value at +60 mV. The data of all the other constructs were also divided by the average current amplitude of the wild-type at +60 mV of the respective recording day. (**C**) Current amplitudes analyzed at +40 mV and normalized to KCNQ1^WT^ (100%). All the data, also that of the wild-type recordings, were divided by the average current amplitude of the wild-type at +40 mV of the respective recording day. Numbers of oocytes recorded are indicated within the bar graphs. Values are expressed as means ± S.E.M. Error bars represent S.E.M. values. Significance was assessed using two tailed Student’s t-test. Asterisks indicate significance: ***, *p* < 0.001.

**Figure 3 ijms-22-01112-f003:**
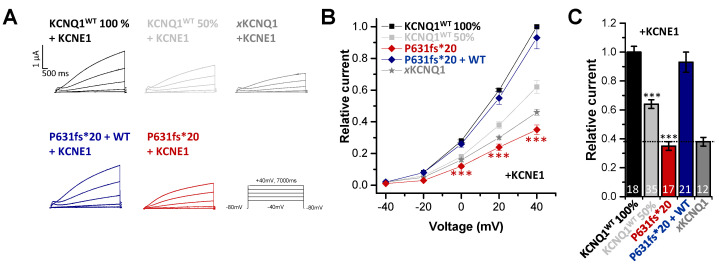
*Functional effects of the KCNQ1^P631fs*20^ mutant co-expressed with KCNE1.* (**A**) Representative current traces of oocytes injected with KCNQ1^WT^ (8.1 ng/oocyte), KCNQ1^P631fs*20^ (8.1 ng/oocyte), KCNQ1^WT^ 50% (4 ng/oocyte) or KCNQ1^WT^ plus KCNQ1^P631fs*20^ (4 ng/oocyte each), all together with 0.1 ng KCNE1 cRNA or only KCNE1 cRNA (0.1 ng/oocyte), to obtain *Xenopus* (*x*)KCNQ1/KCNE1 (*xI*_Ks_) channels. Voltage was stepped from -40 to +40 mV in 20 mV increments lasting 7000 ms, from a holding potential of −80 mV. (**B**) Current voltage relationships obtained by blotting the current at the end of each voltage step for each voltage applied normalized to KCNQ1^WT^ plus KCNE1. In order to obtain the current-voltage relationship (I/V curve), all wild-type recordings were normalized to the value at +60 mV. The data of all the other constructs were also divided by the average current amplitude of the wild-type at +60 mV of the respective recording day. (**C**) Current amplitudes analyzed at +40 mV and normalized to KCNQ1^WT^ plus KCNE1. All the data, also that of the wild-type recordings, were divided by the average current amplitude of the wild-type at +40 mV of the respective recording day. (**A**–**C**) All data are presented without subtraction of *x*KCNQ1 background currents. Numbers of oocytes recorded are indicated within the bar graphs. Values are expressed as means ± S.E.M. Error bars represent S.E.M. values. Significance was assessed using two tailed Student’s *t*-test. Asterisks indicate significance: ***, *p* < 0.001.

**Figure 4 ijms-22-01112-f004:**
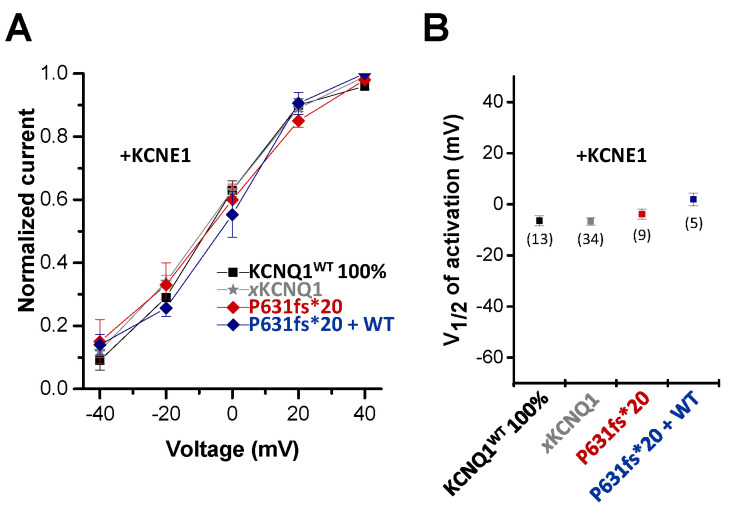
*Voltage-dependence of activation of the KCNQ1^P631fs*20^ mutant co-expressed with KCNE1.* (**A**) Voltage-dependence of activation for KCNQ1+KCNE1, *x*KCNQ1+KCNE1, KCNQ1^P631fs*20^+KCNE1 and KCNQ1^WT^/KCNQ1^P631fs*20^+KCNE1. Recordings were performed with the protocol as described in Figure 3. The tail currents recorded after the 7000 ms pulse were normalized to the respective maximal tail current of each recording to obtain the conductance/voltage (G/V) curves. (**B**) Normalized tail currents were fitted to a Boltzmann function. The voltage of half-maximal activation (V_½_) of KCNQ1+KCNE1, *x*KCNQ1+KCNE1, KCNQ1^P631fs*20^+KCNE1 and KCNQ1^WT^/KCNQ1^P631fs*20^+KCNE1 are illustrated, together with the numbers of oocytes analyzed.

**Figure 5 ijms-22-01112-f005:**
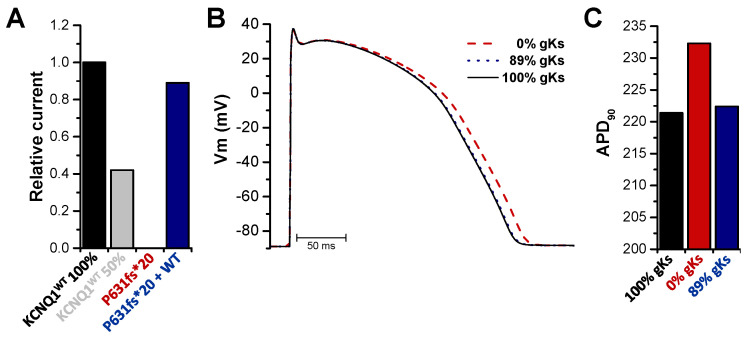
*Action potential modelling with the homozygous or heterozygous KCNQ1^P631fs*20^ mutation in the I_Ks_ channel complex.* (**A**) Mean current amplitudes of 100% KCNQ1^WT^ injected together with KCNE1, 50% KCNQ1^WT^ injected together with KCNE1, 100% KCNQ1^P631fs*20^ injected together with KCNE1 or 50% KCNQ1^WT^ plus 50% KCNQ1^P631fs*20^ injected together with KCNE1. For all data, the endogenous *xI*_Ks_ current recorded by injection of KCNE1 alone was subtracted. Data were normalized to KCNQ1^WT^ 100%. (**B**) Action potential simulations using the O’Hara-Rudy ventricular cell model. Conductivity of the *I*_Ks_ channel was set to 100% for wild-type (black), 0% for the homozygous (red, dashed line) and 89% for the heterozygous state (blue, dotted line) as calculated in (**A**). (**C**) APD at 90% repolarization (APD_90_) was evaluated for each configuration after the simulations achieved a steady state.

## Data Availability

Data is contained within the article. Raw data are available from the corresponding author on reasonable request.

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
