# Peer review of "Molecular Mechanism of Autosomal Recessive Long QT-Syndrome 1 without Deafness"

_ijms, 2021, doi:10.3390/ijms22031112_

Round 1

Reviewer 1 Report

KCNQ1-KCNE1 channel complexes generate the slow delayed rectifier potassium current (IKs) in the heart, which contributes to the repolarization of the cardiac action potential. Loss-of-function mutations in KCNQ1 and KCNE1 genes can cause type 1 and type 5 long QT syndrome, respectively, which predisposes carriers to ventricular cardiac arrhythmia.

Oertli and coworkers report the identification and experimental characterization of a KCNQ1 variant, c.1892-1893insC (p.P631fs*20), found in a patient with an autosomal recessive LQTS inheritance trait. The P631fs*20 variant leads to a C-terminally truncated channel protein due to a shift in the DNA reading frame and a premature stop codon at amino acid position 651.

Using voltage-clamp recordings of KCNQ1 in Xenopus laevis oocytes, a loss-of-function phenotype of channels formed by the P631fs*20 variant is confirmed. Concurrent expression of KCNQ1 WT and P631fs*20 variant leads to channels with an intermediate current amplitude, which is reduced compared to WT but higher than for homomeric KCNQ1P631fs*20 channels. Co-expression with KCNE1 is shown to restore function of heteromeric KCNQ1WT-KCNQ1P631fs*20 channel complexes but not of homomeric mutant KCNQ1P631fs*20 channels. Finally, computational modeling based on experimentally determined current data predicts an increased action potential duration and QT prolongation for homozygous mutation carriers, which is not observed under heterozygous conditions.

These are interesting findings, which highlight a role of the KCNE1 auxiliary subunit in regulating KCNQ1 biogenesis as well as of KCNQ1 protein mistrafficking as common disease mechanism leading to LQTS. The manuscript is well written and the experimental procedures and data analyses are technically sound. I can recommend publication of the manuscript in the International Journal of Molecular Sciences. I kindly ask the authors to address the following minor comments/points.

Minor comments:

1) The authors suggest that the impaired function of KCNQ1P631fs*20 is due to protein trafficking defects, as shown in an earlier study by Sato et al., and that association with KCNE1 can rescue trafficking in case of heteromeric channel complexes formed by KCNQ1 WT and P631fs*20. Are any cell surface expression or trafficking data for P631fs*20 and wildtype KCNQ1 (with and without KCNE1) available in order to confirm this hypothesis? What is known about the underlying molecular mechanism by which KCNE1 restores trafficking? It appears that the effect on the P631fs*30 subunit is indirect and that rescue of trafficking depends on the presence of wildtype KCNQ1 subunits in the formed channel complex.

2) Can the authors explain if under heterozygous expression conditions a stochastic assembly of all possible combinations of homo- and heteromeric KCNQ1WT and KCNQ1P631fs*20 channels can be assumed?

3) Can the authors clarify if the current traces shown in Fig 3A and the I-V relations in Fig 3B are corrected for Xenopus KCNQ1 background current or if data are shown without subtraction of background current? This is not absolutely clear given the legend to Fig 3.

4) Under Methods, can the authors provide some of the simulation parameters used for action potential modeling?

Other points:

Line 68-70: “Besides in the heart, KCNQ1 is expressed in several tissues including kidney, rectum, pancreas, lung, placenta, stomach, adrenal gland, thyroid gland, colon and the stria vascularis of the cochlear duct.” Please add references where the aforementioned information can be found.

Line 97:Lastly the autosomal …” should be “Lastly, the autosomal”. Missing comma.

Line 125-126: “An insertion of cytosine …” should be “An insertion of cytidine” or “An insertion of deoxycytidine …”. Cytosine is one of the nucleobases, but what is inserted is a nucleotide.

Line 130-132: “The variant was absent from the Exome Variant Server (EVS) and not found by the ExAC browser. Referring to the Protein Variation Effect Analyzer (PROVEAN)-score P631fs*20 was predicted to be deleterious.” Please provide references for these methods.

Line 134:Additionally one paternal uncle …” should be “Additionally, one paternal uncle …”. Missing comma.

Line 143-145: “Partial nucleotide and amino acid sequence […] for KCNQ1WT and the KCNQ1P631fs*20 mutation.” I think, it is better to use “KCNQ1P631fs*20 variant” in this sentence.

Line 145: “The cysteine insertion …” should be “The cytidine insertion …”.

Line 148: “… by a red cycle” should be “… by a red circle”.

Line 164, 166: “resampling” should be “resembling” or “reflecting”.

Line 168, 199: “… lead to …” should be “… led to …”. Simple Past.

Line 170: “… compared to the KCNQ1WT (Figure 1C)” should be “… compared to the KCNQ1WT (Figure 2C)”

Line 178: “Voltage was stepped from -60 and +60 mv in 20 mV-steps …” should be “Voltage was stepped from -60 mV to +60 mV in 20 mV-steps …”.

Line 216: “Voltage was stepped from -40 and +40 mv in 20 mV-steps …” should be “Voltage was stepped from -40 mV to +40 mV in 20 mV-steps …”.

Line 246: “… 100% for wild-type (black) 0% for the homozygous …” should be “… 100% for wild-type (black), 0% for the homozygous …”. Missing comma.

Line 289: “idenftied” should be “identified”.

Line 289: “… we speculate that that KCNE1” should be “… we speculate that KCNE1”. Delete one “that”.

Line 302:Thus the amount of …” should be “Thus, the amount of …”. Missing comma.

Line 347: “… lead to a current reduction of …” should be “… led to a current reduction of …”. Simple Past.

Line 361: “According to Goldberg et al. [32] we considered …” should be “According to Goldberg et al. [32], we considered …”. Missing comma.

Reviewer 2 Report

In this manuscript the authors provide an electrophysiological characterization of a mutant of KCNQ1 (KCNQ1P636fs*20 ) carrying a mutation responsible for a loss of function. They first established the pedigree of the family showing that members heterozygous for the mutation are asymptomatic while the IP homozygous developed LQTS. They thus show that in the absence of KCNE1 (the beta subunit of KCNQ1) the expression of KCNQ1P636fs*20 induces a much smaller potassium current than the expression of KCNQ1. This reduction of the potassium current is also observed when KCNQ1P636fs*20 is co-expressed with KCNQ1, mimicking the heterologous situation. In contrast in the presence of KCNE1, the co-expression of KCNQ1P636fs*20 with KCNQ1 give rise to a potassium current similar to the one measured with only KCNQ1. However the expression of only KCNQ1P636fs*20 together with KCNE1 results in an almost complete loss of potassium current.

The authors then analyze these results in the light of the previous phenotypic description of this KCNQ1 mutation. They propose that in contrast of the homomeric KCNQ1P636fs*20  form of the  channel the heterotetrametric form composed of both wt KCNQ1 and KCNQ1P636fs*20 is targeted to the plasma membrane and able to carry a sufficient potassium current.

Although the message is straight forward, several points remain to be clarified in order to improve the manuscript.

  • Line 135. The sister of the IP is described as heterologous for the mutation but is noted as N in the pedigree of figure 1.
  • Figure 1 legend:
  • line 154: “ -/- homozygous mutation carrier” but the IP is marked +/+ in the pedigree.
  • Line 157: “cause of death is indicated below”. This does not appear in the pedigree
  • Figure 2: since the electrophysiological characterization is the major objective the study it would be interesting to show the tail current amplitude as function of voltage in order to investigate the influence of the mutation on the real voltage dependency.
  • Figure 2: current normalization should better described (line 180). If the currents are normalized to the maximum amplitude of the KCNQ1 current, why no error bares for the different potentials for the KCNQ1wt current (as shown in figure 2C)? In fact it would be more interesting to show the real current amplitude as function of voltage instead of normalized current.
  • Legend of figure 2 there is no * nor ** (line 183).
  • The upper remarks are applicable to the results presented on figure 3.
